# Evidence for Genetic Causal Association Between the Gut Microbiome, Derived Metabolites, and Age-Related Macular Degeneration: A Mediation Mendelian Randomization Analysis

**DOI:** 10.3390/biomedicines13030639

**Published:** 2025-03-05

**Authors:** Pinghui Wei, Shan Gao, Guoge Han

**Affiliations:** 1Tianjin Eye Hospital, Tianjin Key Lab of Ophthalmology and Visual Science, Tianjin 300020, China; xmsswph@163.com (P.W.); 2113936@mail.nankai.edu.cn (S.G.); 2Nankai University Eye Institute, Nankai University Affiliated Eye Hospital, Nankai University, Tianjin 300071, China; 3Clinical College of Ophthalmology, Tianjin Medical University, Tianjin 300020, China; 4School of Medicine, Nankai University, Tianjin 300071, China

**Keywords:** gut microbiota, mediation analysis, Mendelian randomization, metabolites, age-related macular degeneration

## Abstract

**Background/Objectives**: Despite substantial research, the causal relationships between gut microbiota (GM) and age-related macular degeneration (AMD) remain unclear. We aimed to explore these causal associations using Mendelian randomization (MR) and elucidate the potential mechanisms mediated by blood metabolites. **Methods**: We utilized the 211 GM dataset (n = 18,340) provided by the MiBioGen consortium. AMD outcome data were sourced from the MRC Integrated Epidemiology Unit (IEU) OpenGWAS Project. We performed bidirectional MR, two mediation analyses, and two-step MR to assess the causal links between GM and different stages of AMD (early, dry, and wet). **Results**: Our findings indicate that the *Bacteroidales S24.7* group and genus *Dorea* are associated with an increased risk of early AMD, while *Ruminococcaceae UCG011* and *Parasutterella* are linked to a higher risk of dry AMD. Conversely, *Lachnospiraceae UCG004* and *Anaerotruncus* are protective against dry AMD. In the case of wet AMD, *Intestinimonas* and *Sellimonas* increase risk, whereas *Anaerotruncus* and *Rikenellaceae RC9* reduce it. Additionally, various blood metabolites were implicated: valine, arabinose, creatine, lysine, alanine, and apolipoprotein A1 were associated with early AMD; glutamine and hyodeoxycholate—with a reduced risk of dry AMD; and androsterone sulfate, epiandrosterone sulfate, and lipopolysaccharide—with a reduced risk of wet AMD. Notably, the association between family *Oxalobacteraceae* and early AMD was mediated by valine, accounting for 19.1% of the association. **Conclusions**: This study establishes causal links between specific gut microbiota and AMD, mediated by blood metabolites, thereby enhancing our understanding of the gut–retina axis in AMD pathophysiology.

## 1. Introduction

Age-related macular degeneration (AMD) is a significant global health concern and is responsible for 8.7% of all cases of blindness worldwide. It is estimated that its incidence will reach 288 million by 2040, significantly increasing the global health and economic burden of the disease [1]. Although AMD is believed to be driven by the complex interplay between genetic, environmental, and metabolic factors [2], the precise etiology is unknown, and a better understanding of the underlying pathophysiological mechanisms is crucial for reducing the incidence of AMD and facilitating the development of novel treatments.

Despite advances made over the past decade in the use of intravitreal injections of anti-vascular endothelial growth factor drugs to treat wet AMD, a certain proportion of patients continue to experience progressive central vision loss [3]. Moreover, no effective treatment options are currently available for those with early-stage and dry AMD, leaving many patients with limited therapeutic options. Therefore, those in the scientific community have continued to explore alternative pathophysiological mechanisms that could lead to the development of new preventive and treatment strategies, including those that focus on the role of the gut microbiota (GM) in related processes.

The GM plays a pivotal role in nutrient metabolism, the regulation of inflammation, and immune system functionality [4]. A growing body of evidence suggests that the GM is associated with several eye diseases, including diabetic retinopathy, uveitis, and glaucoma [5,6]. Additionally, some studies have suggested that obesity-associated GM can drive abnormal choroidal neovascularization within retinal tissue, thereby supporting the concept of a “gut–retina axis” that contributes to the development of eye disease [7]. Nevertheless, the precise relationship between GM taxa, GM-derived metabolites, and the stages of AMD remains unclear.

Mendelian randomization (MR) is a robust methodology that employs genetic variants as instrumental variables (IVs) to facilitate causal inferences in observational studies [8]. By leveraging genetic variants associated with the GM composition and the levels of GM-related metabolites, it is possible to infer their potential causal relationships with AMD while simultaneously minimizing the effects of confounding factors [9]. For example, the two-sample MR method offers greater statistical power than other methods in identifying causal effects between “exposure factors” and “outcome” variables by leveraging published summary estimates from large-scale genome-wide association studies (GWAS). The increasing availability of large-scale summary statistics has enabled the analysis of relationships between the GM or derived metabolites and AMD, thereby enhancing the statistical power of two-sample MR analyses.

Despite growing evidence linking the GM to eye diseases, the causal relationships between specific GM taxa and metabolites with different stages of AMD remain poorly understood. This study aims to address this gap by employing Mendelian randomization to explore these causal associations.

## 2. Materials and Methods

### 2.1. Study Design

Figure 1 illustrates the study design. First, bidirectional MR analyses were conducted to explore the potential causal relationships between the GM and AMD. Subsequently, two-step two-sample MR (TSMR) was performed using public datasets to identify genome-wide associations between the GM and AMD based on the levels of metabolites in blood samples. More specifically, in step one, TSMR analysis was used to evaluate the causal pathway from the GM to metabolites; in step two, the potential mediators of the GM-related metabolites on AMD were investigated. The study was conducted in accordance with the principles of the Declaration of Helsinki. Ethics approval and consent were not required due to the use of anonymized, publicly accessible datasets in this study.

### 2.2. Genetic Instruments for the GM

The present study utilized the GM dataset provided by the MiBioGen consortium, which comprises 16S ribosomal ribonucleic acid (rRNA) gene sequencing data from 18,340 individuals of European ancestry from 24 cohorts, along with the corresponding information pertaining to their genotype [10]. This dataset encompasses a total of 211 taxonomic units spanning from the phylum to genus levels, in addition to information on 122,110 sites of genomic variation. To obtain more accurate results, data for three unknown families and 12 unknown genera of the GM taxa were removed. Leveraging the GWAS conducted by the MiBioGen consortium, instrumental variables (IVs) representing five levels of GM taxonomic units (phylum, class, order, family, and genus) were identified. To satisfy the three fundamental assumptions of MR, all single nucleotide polymorphisms (SNPs) were subjected to rigorous quality checks. SNPs falling within the gene locus range associated with GM taxonomic units (with a significance threshold of *p* < 1 × 10^−5^) were selected to serve as the IVs. Furthermore, linkage disequilibrium (LD) analysis (with criteria of an R^2^ < 0.001 and a clustering distance of 10,000 kb) was conducted to ensure adherence to the MR assumptions.

### 2.3. Genetic Instruments for GM-Related Metabolites

To further investigate the correlations between certain GM-related metabolites and the occurrence of AMD, summary-level data encompassing 486 metabolites from a human metabolome-wide association study involving 7824 individuals of European ancestry were analyzed [11]. Subsequently, 79 metabolites derived from the gut microbiota were identified using the Human Metabolome Database (HMDB) [12]. In addition, apolipoprotein A1 levels and serum lipopolysaccharide activity were utilized to analyze the potential causal association with AMD. Selection of genetic IVs was performed with a genome-wide significance threshold (*p* < 5 × 10^−6^). Pairwise LD between the SNPs was assessed to identify the SNPs exhibiting R^2^ values below 0.001 within a clumping window spanning 10,000 kb. Additionally, the IVs were evaluated based on their F-values, and only those with values that exceeded 10 were included in the analysis. This stringent criterion was utilized to mitigate any potential bias stemming from weak IVs.

### 2.4. Genetic Instruments for AMD Subtypes

The AMD outcome data used in this study were sourced from the MRC Integrated Epidemiology Unit (IEU) OpenGWAS Project. The data analyzed in this study for three distinct subtypes of AMD, namely, early, dry, and wet AMD, were obtained from individuals of European ancestry. Early AMD is characterized by yellowish accumulations of extracellular material of varying sizes between Bruch’s membrane and the retinal pigment epithelium (RPE) or between the RPE and photoreceptors (known as drusen or subretinal drusenoid deposits, respectively). The data analyzed in this study were derived from a substantial cohort comprising 14,034 patients with AMD and 91,214 controls from 11 diverse sources, including prominent entities such as the International AMD Genomics Consortium (IAMDGC) and the United Kingdom Biobank (UKBB) [13]. Moreover, this study also incorporated data on the dry and wet AMD subtypes; the GWAS data for dry AMD were derived from 2469 cases and 206,221 controls, whereas the data for wet AMD were derived from 2114 cases and 206,601 controls.

### 2.5. Statistical Analysis

To investigate potential causal relationships between GM taxa, GM-derived metabolites, and AMD phenotypes, MR analyses were conducted using the TwoSampleMR, ggplot2, and Mendelian Randomization packages within the R statistical software platform (version 4.3.1). In our analysis, for features that had only one instrumental variable (IV), the Wald ratio method was used to estimate the causal relationship between an exposure factor and outcome variable. When dealing with features involving multiple IVs, we utilized five well-known MR methods, namely, the inverse-variance weighted (IVW) method, the weighted mode method, the MR-Egger regression, the weighted median estimator (WME), and MR-PRESSO. It has been reported in the literature that when all genetic variants are valid instrumental variables, the IVW method exhibits a slightly greater statistical power compared to the other four methods [14]. Therefore, for features with more than one IV, our primary results were mainly based on the IVW method, while the other four methods were used as supplementary analyses to provide a more comprehensive understanding of the results. The primary analyses utilized the IVW method to estimate causal effects, with the significance threshold set at *p* < 0.05. More specifically, we established a multiple-testing significance threshold for each feature level, including phylum, class, order, family, and genus. This threshold was defined as *p* < 0.05/n, where n represents the effective number of independent bacterial taxa at the corresponding taxonomic level. In particular, we applied the Bonferroni approach to correct for multiple hypothesis testing, setting the significance threshold at Bonferroni-corrected *p*-values < 0.05. Moreover, associations with *p* < 0.05 that did not meet the Bonferroni-corrected threshold were reported as potentially indicative of an association. Odds ratio (OR) and the corresponding 95% confidence intervals (CIs) were calculated to elucidate the direction (positive or negative) of causal effects. Notably, when the IVW method was used, the regression model intercept was constrained to zero.

Sensitivity analyses utilizing the MR-Egger method, which incorporates an intercept term (*p* < 0.05), were conducted to address potential biases arising from directional pleiotropy. Additionally, the MR pleiotropy residual sum and outlier (MR-PRESSO) method was used to identify outliers and assess the overall heterogeneity in the MR estimates. Furthermore, Cochran’s Q test was used to scrutinize the degree of heterogeneity among the SNPs; in instances where heterogeneity was detected, particular attention was paid to the results obtained from the IVW method utilizing multiplicative random effects. The code used is publicly available at https://github.com/xmsswph/Gut-Microbiome-Derived-Metabolites-and-AMD, accessed on 17 February 2025.

In order to estimate the mediation effect of GM-related metabolites on the association between the GM and AMD, two-step MR was performed. Specifically, (1) in the first step, a univariable MR was carried out to detect the causal effect of the GM on the metabolites (β1); (2) in the second step, the metabolites were considered as the exposure variable to detect the causal effect on AMD (β2); and (3) in the third step, the causal effect of the GM on AMD (βall) was estimated. Thus, the mediation effect could be calculated as β1 × β2, the direct effect as βall − β1 × β2, and the mediation proportion could be calculated as β1 × β2/βall.

To evaluate the influence of individual SNPs on the overall MR findings, a comprehensive leave-one-out analysis was performed; this iterative process involved the systematic exclusion of genetic variants from each chromosome and the recalculation of the MR-IVW estimates at each step.

## 3. Results

### 3.1. IVs for the Exposure Factors Included in the Analyses

The number of SNPs used as IVs for the analyses can be found in the Appendix A.

### 3.2. Causal Relationships Between the GM and AMD Subtypes

In the early AMD group, the MR results revealed that certain microbial taxa promote disease progression (Figure 2B), including the family *Bacteroidales S24.7* group (OR = 1.19, 95% CI = 1.06–1.34, *p* = 4.50 × 10^−3^), the family *Oxalobacteraceae* (OR = 1.09, 95% CI = 1.01–1.19, *p* = 3.69 × 10^−2^), the genus *Dorea* (OR = 1.23, 95% CI = 1.01–1.49, *p* = 3.81 × 10^−2^), and the genus *Parasutterella* (OR = 1.11, 95% CI = 1.01–1.24, *p* = 4.66 × 10^−2^). Conversely, other taxa were found to reduce the risk of early AMD, including the genus *Lachnospiraceae UCG004* (OR = 0.84, 95% CI = 0.72–0.98, *p* = 2.25 × 10^−2^), the phylum *Verrucomicrobia* (OR = 0.87, 95% CI = 0.76–0.99, *p* = 3.83 × 10^−2^), the genus *Eubacterium oxidoreducens* group (OR = 0.86, 95% CI = 0.75–0.99, *p* = 3.89 × 10^−2^), and the genus *Ruminiclostridium 5* (OR = 0.86, 95% CI = 0.74–0.99, *p* = 4.66 × 10^−2^).

In the case of dry AMD, the MR results indicated that certain microbial taxa increased the risk of the disease (Figure 2B), such as the genera *Ruminococcaceae UCG011* (OR = 1.29, 95% CI = 1.06–1.58, *p* = 1.27 × 10^−2^), *Parasutterella* (OR = 1.25, 95% CI = 1.00–1.56, *p* = 4.51 × 10^−2^), *Bilophila* (OR = 1.29, 95% CI = 1.00–1.67, *p* = 4.80 × 10^−2^), and *Faecalibacterium* (OR = 1.35, 95% CI = 1.04–1.75, *p* = 2.51 × 10^−2^). Conversely, other taxa were associated with a decreased risk of dry AMD, including the genera *Holdemanella* (OR = 0.75, 95% CI = 0.62–0.91, *p* = 3.94 × 10^−3^), *Phascolarctobacterium* (OR = 0.67, 95% CI = 0.51–0.89, *p* =6.44 × 10^−3^), *Anaerotruncus* (OR = 0.64, 95% CI = 0.46–0.88, *p* = 6.66 × 10^−3^), *Bacteroides* (OR = 0.67, 95% CI = 0.48–0.93, *p* = 1.71 × 10^−2^), and *Slackia* (OR = 0.76, 95% CI = 0.60–0.98, *p* =3.43 × 10^−2^), and the family *Bacteroidaceae* (OR = 0.67, 95% CI = 0.48–0.93, *p* =1.71 × 10^−2^).

In the case of wet AMD, certain GM taxa were found to increase the risk of disease onset (Figure 2B), including the genera *Intestinimonas* (OR = 1.28, 95% CI = 1.01–1.63, *p* = 3.99 × 10^−2^) and *Sellimonas* (OR = 1.21, 95% CI = 1.00–1.47, *p* = 4.99 × 10^−2^), whereas others were associated with a decreased risk of disease onset, including the genera *Anaerotruncus* (OR = 0.65, 95% CI = 0.47–0.89, *p* = 7.90 × 10^−3^) and the *Rikenellaceae* RC9 gut group (OR = 0.83, 95% CI = 0.69–0.99, *p* = 3.59 × 10^−2^).

The causal effects of AMD subtypes on GM taxa were also assessed. Only the genus *Romboutsia* decreased after early AMD (OR = 0.63, 95% CI = 0.51–0.78, *p* = 2.11 × 10^−5^), whereas the genera *Romboutsia* (OR = 0.78, 95% CI = 0.70–0.88, P =2.11 × 10^−5^) and the *Eubacterium hallii* group (OR = 0.77, 95% CI = 0.68–0.87, *p* =3.21 × 10^−5^) decreased after wet AMD. However, no bidirectional causal relationships were identified between dry AMD and any GM taxon.

### 3.3. Mediation Analysis of the Effects of GM-Related Metabolites on AMD

In the two-sample MR analyses (Figure 3), lower levels of valine (OR = 0.21, 95% CI = 0.06–0.67, *p* = 8.31 × 10^−3^), arabinose (OR = 0.76, 95% CI = 0.57–0.99, *p* = 4.81 × 10^−2^), and creatine (OR = 0.58, 95% CI = 0.36–0.93, *p* = 2.40 × 10^−2^) were determined to be negatively associated with early AMD, whereas higher levels of lysine (OR = 3.83, 95% CI = 1.34–10.96, *p* = 1.22 × 10^−2^), alanine (OR = 2.53, 95% CI = 1.37–4.68, *p* = 3.01 × 10^−3^), and apolipoprotein A1 (OR = 1.24, 95% CI = 1.10–1.40, *p* = 4.46 × 10^−4^) were found to be positively linked to early AMD. Subsequently, the two-step MR analyses revealed that only the family *Oxalobacteraceae* promoted the likelihood of early AMD by inhibiting the secretion of valine, with a mediating proportion of 19.1%.

Glutamine (OR = 0.06, 95% CI = 0.01–0.65, *p* = 2.05 × 10^−2^) and hyodeoxycholate (OR = 0.69, 95% CI = 0.48–0.98, *p* = 3.99 × 10^−2^) were both found to be associated with a reduced risk of dry AMD, whereas higher levels of androsterone sulfate (OR = 0.66, 95% CI = 0.51–0.85, *p* = 1.50 × 10^−3^) and epiandrosterone sulfate (OR = 0.47, 95% CI = 0.32–0.71, *p* = 2.27 × 10^−4^) were associated with a decreased risk of wet AMD. However, the serum lipopolysaccharide level was associated with an increased risk of both dry AMD (OR = 1.20, 95% CI = 1.10–1.31, *p* = 2.72 × 10^−5^) and wet AMD (OR = 1.14, 95% CI = 1.04–1.26, *p* = 8.03 × 10^−3^). The two-step MR analysis revealed that only the genus *Slackia* exerted protective effects against the development of dry AMD by upregulating glutamine, with a mediating proportion of 16.6%.

### 3.4. Sensitivity Analyses

The results of the sensitivity analyses are presented in Table 1. No heterogeneity (IVW: *p* ≥ 0.12; MR-Egger: *p* ≥ 0.08) or horizontal pleiotropy (*p* ≥ 0.05) was observed (Table 1) when examining the effects of various GM taxa on the development of early, dry, and wet AMD. When examining the effects of specific metabolites, heterogeneity was detected for the effects of apolipoprotein A1 levels on early AMD (IVW: *p* = 2.72 × 10^−6^; MR-Egger: *p* = 6.50 × 10^−6^), and horizontal pleiotropy was detected for the effects of alanine levels on early AMD (*p* = 0.043) (Table 1). No other heterogeneity (IVW: *p* ≥ 0.21; MR-Egger: *p* ≥ 0.25) or horizontal pleiotropy (*p* ≥ 0.12) was observed between metabolites and AMD groups. No heterogeneity (IVW: *p* ≥ 0.30; MR-Egger: *p* ≥ 0.70) was observed between GM taxa and GM-related metabolites; however, horizontal pleiotropy was detected for the relationship between the genus *Slackia* and glutamine levels (*p* = 0.048).

## 4. Discussion

This large-scale study identified causal associations between eight GM taxa and early AMD, between ten taxa and dry AMD, and between four GM taxa and wet AMD. To investigate possible metabolic mechanisms, GM-related blood metabolites associated with the GM taxa and AMD phenotypes were assessed via mediation analyses involving TSMR, which revealed that the family *Oxalobacteraceae* acted as a risk factor by decreasing circulating valine levels, whereas the genus *Slackia* exerted protective effects against dry AMD by increasing glutamine levels.

The GM, a complex microbial community within the gastrointestinal tract, plays a crucial role in the regulation of human health and disease. The present study revealed an association between the presence of members of the family *Oxalobacteraceae* and the genus *Parasutterella* and an increased risk of early AMD development. Notably, both of these microbial groups belong to the phylum *Proteobacteria*, which has been extensively linked to gut microbial dysbiosis and mitochondria-associated metabolic disorders in previous research [15]. The family *Oxalobacteraceae*, which is renowned for its capacity to degrade oxalic acid, has been linked to an increased risk of type 2 diabetes [16] and non-alcoholic fatty liver disease [17] owing to its increased abundance. Thus, it is postulated that the mechanism through which early AMD is promoted may involve oxalate salt-induced mitochondrial damage and increased oxidative stress. Conversely, *Parasutterella* strains have been observed to exhibit antiglycolytic effects and rely on amino acids such as aspartate, asparagine, and serine to execute metabolic activities, ultimately resulting in succinate production [18]. Additionally, some studies have suggested that *Parasutterella* bacteria may modulate the host immune response or induce inflammation through toxin or metabolite production, which could potentially exacerbate the development and progression of AMD [19].

Furthermore, an increased abundance of the family *Bacteroidales*, a constituent of the phylum *Bacteroidetes*, results in the disruption of the *Firmicutes*/*Bacteroidetes* ratio, a disturbance that is associated with AMD development [20]. *Dorea*, a member of the family *Lachnospiraceae* within the phylum *Firmicutes*, is a gas-producing bacterium that plays a significant role in intestinal barrier formation and exhibits proinflammatory properties. *Dorea* bacteria are known to induce regulatory T cells and inhibit the function of T helper 17 cells, thereby regulating intestinal immune responses and maintaining the integrity of the mucosal barrier [21]. Previous studies have demonstrated a potential association between *Dorea* and various conditions, including fungal keratitis, Sjögren’s syndrome, and dry eye syndrome [22]. Furthermore, this study identified several GM genera that appeared to mitigate the risk of early AMD development, including *Lachnospiraceae* and *Ruminococcaceae*. These core genera of the human GM are likely to participate in carbohydrate metabolism and could exert beneficial effects [23]. The *Lachnospiraceae* family has been extensively studied and proven to have health-promoting functions [24]. A previous study showed that this family of bacteria can produce short-chain fatty acids (SCFAs), such as butyric acid, propionic acid, and acetic acid [25]. As important metabolites of the gut microbiota, SCFAs play a crucial role in maintaining the body’s immune balance and suppressing inflammatory responses [26,27]. In the pathological process of dry AMD, chronic inflammation in the retina is an important pathological feature. Therefore, we speculate that *Lachnospiraceae UCG004* may increase the content of SCFAs, participate in regulating the immune microenvironment of the retina, inhibit the inflammatory response, and finally exert a protective effect against dry AMD. Additionally, *Eubacterium*, another important genus of the human GM, plays a crucial role in nutrient metabolism and the maintenance of intestinal homeostasis, while also exhibiting anti-inflammatory properties, among other beneficial effects [28]. The abundance of these beneficial microbial genera appears to be closely linked to factors associated with early AMD prevention.

Members of the GM harbor a diverse array of genes that are involved in pathways that regulate amino acid metabolism. Studies involving patients with AMD have identified a reduction in the expression levels of various genes associated with these amino acid metabolic pathways [29,30]. The present analyses revealed that both lysine and alanine increased the risk of early AMD development, which is consistent with previous findings. For example, Hou et al. [31] outlined potential metabolic biomarkers and pathways for AMD diagnosis, highlighting the involvement of amino acid metabolism in AMD progression, particularly alanine metabolism and lysine degradation. Furthermore, another previous study identified an association between alanine alleles and the levels of oxidative stress, with certain alleles being expressed at a significantly higher frequency in patients with AMD than in healthy individuals [32]. The dysregulated metabolic demands of RPE cells likely contribute to lipoprotein accumulation and drusen formation, which are important pathological changes at the early stage of AMD. Valine, as a branched-chain amino acid, not only participates in gluconeogenesis, but also plays a key role in fatty acid metabolism-related processes. Some studies have suggested that the catabolism of branched-chain amino acids is essential for facilitating the transport of long-chain fatty acids into the mitochondria, thereby reducing mitochondrial overload and oxidative stress damage [33]. This study provides further evidence that the family *Oxalobacteraceae* may modulate early AMD development by regulating valine levels, potentially serving as a pivotal regulatory node within the gut–retina axis.

In addition to amino acid metabolism, lipid metabolism significantly influences drusen formation. The results of this study indicated that higher apolipoprotein A1 levels are associated with an increased risk of early AMD development. Several biochemical and histological investigations have corroborated the substantial lipid content of drusen, with lipid A being a prominent constituent. Apolipoproteins play a pivotal role in lipid homeostasis, participate in lipid transport and metabolism, influence inflammation, and regulate the immune response [34]. Apolipoprotein A1, a key component of high-density lipoproteins (HDL), is believed to modulate free fatty acid levels in plasma and contributes to HDL and triglyceride-rich lipoprotein metabolism. Some studies have revealed elevated apolipoprotein A1 levels in patients with AMD, suggesting it is involved in disease pathogenesis and could serve as a potential biomarker for AMD [34,35]. Moreover, these insights underscore the therapeutic potential of lipid-lowering drugs in the treatment of AMD [36].

The precise etiology of dry AMD remains unclear, although the prevailing hypothesis implicates damage to the RPE and photoreceptor cells as a key event in its pathogenesis, along with oxidative stress and inflammation [37]. This study identified four GM taxa associated with an increased risk of dry AMD development; among them, the genera *Bilophila* and *Parasutterella* belong to the phylum *Proteobacteria*, which are Gram-negative bacteria. The outer membrane of these bacteria is predominantly composed of lipopolysaccharides (LPS); this is associated with an increased intestinal permeability and drives the development of metabolic disorders and inflammation [38]. In particular, *Bilophila* is an opportunistic pathogen whose increased abundance has been confirmed to contribute to intestinal inflammation. The typical Western diet, which is characterized by low fiber, high sugar, high fat, and high animal protein intake, leads to an increase in the abundance of *Bilophila* in feces [39]. Research has demonstrated that this genus of bacteria can reduce sulfate to hydrogen sulfide gas, which, in turn, can trigger inflammation as well as immune and metabolic disorders [40]. Aging is accompanied by a relative increase in the abundance of the phylum *Proteobacteria* and the genus *Ruminococcaceae* [41], a core member of the human GM involved in carbohydrate metabolism. Previous studies have confirmed the close association between these bacteria and AMD [42], and the present findings, which are consistent with previous laboratory data, provide valuable insights into AMD pathogenesis and progression. In comparison with the levels in the control group, the overabundance of *Ruminococcaceae* in patients was confirmed to be a risk factor for AMD development. Conversely, other GM taxa protected against dry AMD development, such as *Phascolarctobacterium*, a member of the phylum *Firmicutes*; these bacteria can produce short-chain fatty acids [43], including acetate and propionate, which have anti-inflammatory effects. Previous studies that have investigated the relationship between the GM and AMD have also identified the presence of *Slackia* bacteria acts as a protective factor against the disease [44]; however, they failed to differentiate between dry and wet AMD. This MR study confirmed the protective effect of *Slackia* bacteria against the development of dry AMD specifically and further clarified their impact on different types of AMD. Regarding the underlying mechanism, subsequent comprehensive analyses demonstrated that *Slackia* inhibited dry AMD development by facilitating an elevation in plasma glutamine levels. Research has shown that glutamine, a non-essential amino acid, is indispensable for maintaining and sustaining immune competence [45]. Given that immunological processes lie at the core of the pathophysiology of AMD [46,47], this finding is highly relevant. In line with the current results, Kersten et al. previously employed targeted metabolomics techniques and identified glutamine levels as a candidate biomarker for AMD [48]. This implies that glutamine could serve as a potential target for future interventional therapies [48]. Additionally, other studies have carried out metabolomic profiling on the aqueous humor of AMD patients, revealing a reduction in glutamine levels in the aqueous humor of these patients [49]. The disparity in these findings might be attributable to the difference in sampling sites. Owing to the influence of the blood–eye barrier, there remains a contentious issue regarding whether the alteration in plasma glutamine levels can accurately mirror the changes in glutamine within the retina. This matter necessitates further in-depth research for conclusive validation.

Hyodeoxycholate, an important intermediate in bile acid metabolism, reduces the generation of reactive oxygen species, upregulates antioxidant gene expression, and protects RPE cells from inflammation and oxidative damage, thereby inhibiting macular degeneration [50]. In contrast to its lack of association with early AMD, *Bacteroides* was confirmed to exert a protective effect against dry AMD. Research has found that the genus *Bacteroides* is negatively correlated with the complement factor H gene and may play a protective role against AMD by reducing inflammation levels [51]. In addition, this study also confirmed that the *Rikenellaceae RC9* intestinal group exerted a protective effect against the development of wet AMD. The GM influences the energy metabolism characteristics of the host organism. Previous research has suggested that the presence of *Rikenellaceae* may be an indicator of a healthier metabolic state [52], and wet AMD is known to be associated with abnormal energy metabolism in the eye [53]. Other studies have also confirmed the protective effects of *Rikenellaceae* against AMD [44].

The results of this MR study indicated that the genus *Anaerotruncus* exerted a protective effect against the development of both dry and wet AMD, which is consistent with previous findings [42], although prior studies did not differentiate between those two types of AMD. Both *Anaerotruncus* and *Rikenellaceae* belong to the phylum *Firmicutes* and play a pivotal role in maintaining the normal composition and function of the GM required to reduce intestinal inflammation and protect against wet AMD. Conversely, the genus *Sellimonas* has been shown to be associated with increased inflammatory responses in various diseases and could potentially promote wet AMD development by upregulating the expression of inflammatory mediators [54].

The GM can influence physiological processes through the release of various metabolites. The present analyses revealed that epiandrosterone sulfate exerted an inhibitory effect on wet AMD. Epiandrosterone sulfate is converted from dehydroepiandrosterone (DHEA), which serves as a precursor to sex steroids such as testosterone and estradiol, and increased levels of these hormones have been shown to be linked to reduced levels of markers of systemic inflammation [55]. Notably, DHEA administration has been shown to significantly reduce the levels of certain proinflammatory cytokines, including tumor necrosis factor alpha and interleukin 6 in elderly individuals, while also facilitating improvements in visceral fat reduction and glucose tolerance [56]. Conversely, this study revealed that LPS increased individuals’ susceptibility to both dry and wet AMD [38]. LPS, a constituent of Gram-negative bacterial cell walls, acts as an endotoxin that triggers inflammation via the activation of Toll-like receptor 4 on the membranes of host cells [38,57]. In the context of the eye, various cell types exhibit proinflammatory responses upon exposure to LPS, as evidenced by the significant neuroinflammatory reactions that have been observed in LPS-induced retinal explants. Mounting evidence suggests that low-grade inflammation induced by LPS contributes to AMD progression, and the presence of elevated levels of esterified 3-hydroxy fatty acids in blood samples indicates the potential role the LPS burden plays in the early stages of AMD pathophysiology [38,58,59]. Collectively, these findings underscore the cascade of acute and chronic inflammatory processes mediated by LPS, which influence the development and progression of AMD. Researchers have sought to elucidate the mechanisms underlying the association between LPS-induced inflammation and AMD risk, including changes in immune–inflammatory responses, oxidative stress, and RPE aging, and the present analyses provide further information on the mediating effects of certain metabolites.

## 5. Conclusions

In summary, this study comprehensively assessed the causal relationships between GM taxa, GM-related blood metabolites, and AMD subtypes. The GM has been identified as a significant factor that influences the onset and progression of AMD, and these findings highlight the importance of elucidating the underlying mediating metabolic mechanisms linking the GM and AMD development. This study confirmed that certain microbiota, such as the family *Oxalobacteraceae* and the genus *Parasutterella*, are implicated in heightened AMD risk, potentially exacerbating AMD via metabolic dysregulation and the enhanced activation of inflammatory cascades. Conversely, protective effects against AMD were confirmed for certain microbiota, such as *Lachnospiraceae* and *Ruminococcaceae*, which promote ocular health by maintaining gut homeostasis and mitigating inflammatory responses. Finally, the study confirmed that amino acid and lipid metabolism are pivotal drivers of the pathogenesis of AMD, and these processes are modulated by certain microbiota. For instance, the family *Oxalobacteraceae* acts as a risk factor by decreasing circulating valine levels, whereas the genus *Slackia* exerts protective effects against the development of dry AMD by increasing glutamine levels. These findings offer crucial insights for future AMD prevention and treatment strategies that target the GM or modulate specific GM-related metabolites. While our current study focuses on the overall influence of the GM on AMD, future research should be directed towards identifying the specific bacteria involved and exploring their potential roles in AMD pathogenesis.

## Figures and Tables

**Figure 1 biomedicines-13-00639-f001:**
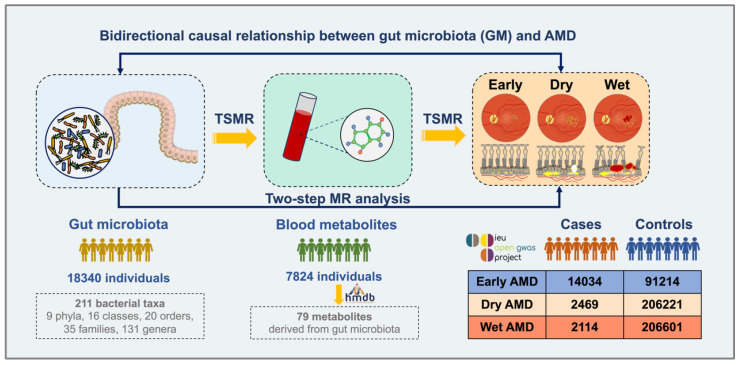
Overview of the Mendelian randomization (MR) framework used to investigate the causal effect of the gut microbiota, blood metabolites derived from the gut microbiota, and age-related macular degeneration (AMD).

**Figure 2 biomedicines-13-00639-f002:**
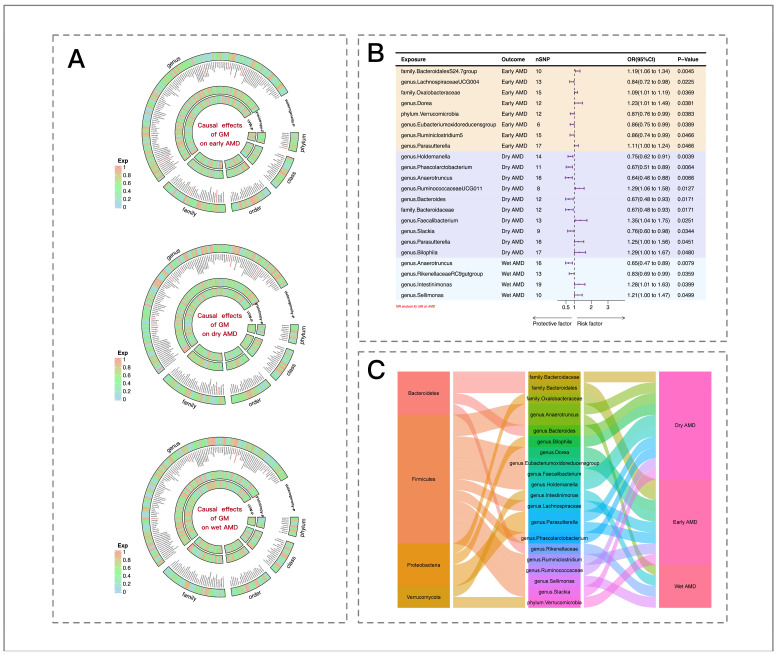
Causal relationship between the gut microbiota (GM) and age-related macular degeneration (AMD). (**A**) Causal and sensitivity analyses were conducted for each gut microbiome taxon across five levels in relation to AMD. The outer circle represents the *p*-value of the heterogeneity test (Cochran’s Q), followed by the GM taxon name, the *p*-value of the pleiotropy test (MR-Egger regression), and the *p*-value based on the IVW results (significant results highlighted in red). Color coding for the *p*-values is based on an RGB color scale (*p* = 0, #ACD6EC; *p* = 0.5, #90ee90; *p* = 1, #F5A899). (**B**) The Mendelian randomization (MR) results reveal the causal relationship between the GM and AMD. With OR = 1 as the reference line, the left side indicates that this GM is a protective factor for AMD, while the right side indicates that this GM is a risk factor for AMD. (**C**) The Sankey diagram illustrates the relationship between the GM and AMD phenotypes. The leftmost side represents the phylum where the gut microbiota comes from, the middle represents the GM, and the rightmost side represents AMD phenotypes.

**Figure 3 biomedicines-13-00639-f003:**
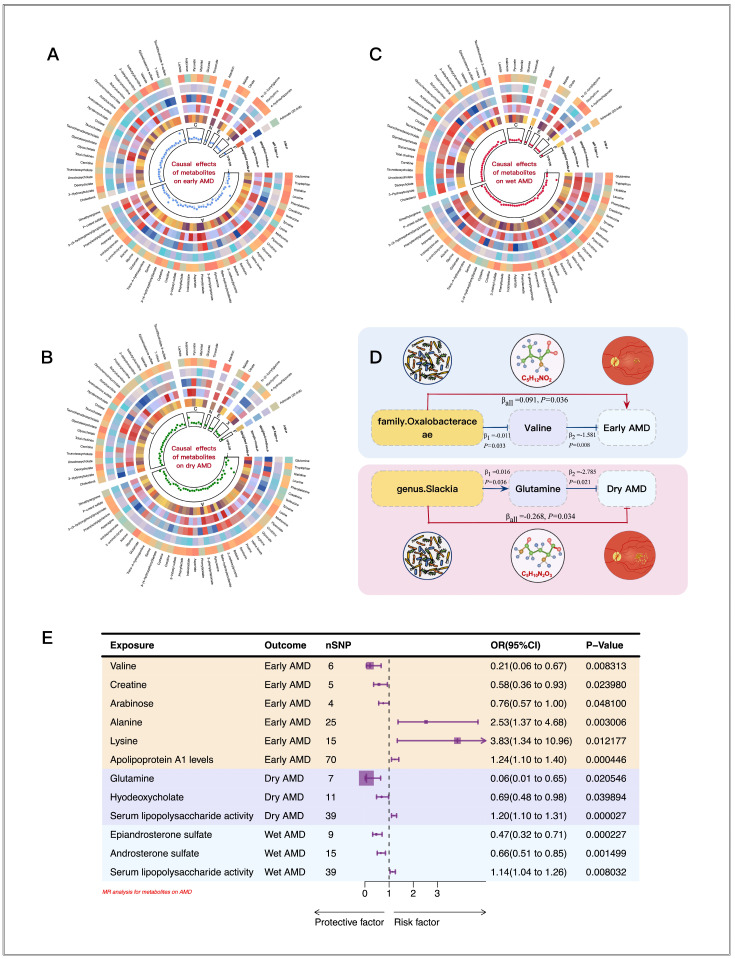
Causal analysis of gut microbiome (GM)-derived metabolites and AMD based on Mendelian randomization (MR) analyses. (**A**–**C**) Results of early, dry, and wet AMD, respectively. From outside to inside, the *p*-values of inverse-variance weighted (IVW), MR-Egger, weighted median, simple mode, and weighted mode are represented, respectively. The odds ratio (OR) value of the IVW method is represented in the innermost side. Groups A, L, C, N, E, X, and F represent amino acids, lipids, carbohydrates, nucleotides, energy, xenobiotics, and fatty acids, respectively. (**D**) The mediation effect of “gut microbiota–blood metabolites–AMD” in two-step Mendelian randomization. (**E**) The Mendelian randomization (MR) results reveal the causal relationship between the GM and AMD. With OR = 1 as the reference line, the left side indicates that this metabolite is a protective factor for AMD, while the right side indicates that this metabolite is a risk factor for AMD. CI indicates confidence intervals.

**Table 1 biomedicines-13-00639-t001:** Heterogeneity and pleiotropy analyses between the exposure and the outcome based on the MR results.

Exposure	Outcome	Heterogeneity				Pleiotropy		
		IVW		MR−Egger				
**Taxa**	**AMD**	**Q**	** *p* **	**Q**	** *p* **	**Intercept**	**Se**	** *p* **
Genus *Lachnospiraceae* *UCG004*	Early	13.34	0.345	13.34	0.272	−0.0004	0.021	0.985
Genus *Dorea*	Early	16.68	0.118	16.67	0.082	−0.0016	0.021	0.940
Genus *Eubacterium* *oxidoreducens* group	Early	2.27	0.811	1.76	0.780	−0.0187	0.026	0.515
Family *Bacteroidales S24.7* group	Early	5.55	0.784	2.18	0.975	−0.0503	0.027	0.104
Family *Oxalobacteraceae*	Early	16.07	0.309	15.94	0.252	−0.0078	0.024	0.754
Phylum *Verrucomicrobia*	Early	9.55	0.571	9.46	0.489	−0.0049	0.016	0.771
Genus *Parasutterella*	Early	15.55	0.484	14.18	0.511	−0.0155	0.013	0.260
Genus *Bilophila*	Dry	15.74	0.472	15.71	0.402	0.0062	0.040	0.878
Genus *Holdemanella*	Dry	13.86	0.384	12.53	0.404	−0.0370	0.033	0.281
Genus *Bacteroides*	Dry	10.78	0.462	10.76	0.377	−0.0093	0.061	0.881
Genus *Ruminococcaceae UCG011*	Dry	8.52	0.289	8.48	0.205	−0.0124	0.074	0.872
Family *Bacteroidaceae*	Dry	10.78	0.462	10.76	0.377	−0.0093	0.061	0.881
Genus *Anaerotruncus*	Dry	18.27	0.249	17.40	0.236	0.0284	0.034	0.417
Genus *Phascolarcto bacterium*	Dry	6.09	0.808	5.24	0.813	−0.0450	0.049	0.382
Genus *Slackia*	Dry	0.67	1.000	0.67	0.999	−0.0024	0.063	0.970
Genus *Parasutterella*	Dry	14.19	0.511	13.16	0.514	0.0266	0.026	0.327
Genus *Faecalibacterium*	Dry	5.27	0.948	5.05	0.929	0.0123	0.026	0.647
Genus *Intestinimonas*	Wet	15.24	0.645	15.24	0.578	0.0056	0.033	0.866
Genus *Sellimonas*	Wet	12.09	0.208	7.18	0.517	0.1332	0.071	0.086
Genus *Rikenellaceae RC9* gut group	Wet	15.14	0.234	11.43	0.408	0.1654	0.075	0.057
Genus *Anaerotruncus*	Wet	7.41	0.945	7.38	0.919	−0.0018	0.028	0.949
**Metabolites**	**AMD**	**Q**	** *p* **	**Q**	** *p* **	**Intercept**	**Se**	** *p* **
Apolipoprotein A1 levels	Early	136.03	2.72 × 10^−6^ *	131.27	6.50 × 10^−6^ *	−0.0082	0.005	0.121
Arabinose	Early	0.88	0.830	0.45	0.797	0.0106	0.016	0.580
Lysine	Early	13.79	0.466	12.68	0.473	−0.0263	0.025	0.313
Valine	Early	4.77	0.444	1.03	0.906	0.0194	0.010	0.125
Creatine	Early	3.44	0.487	0.33	0.953	−0.0342	0.019	0.176
Alanine	Early	16.02	0.887	11.42	0.978	0.0226	0.011	0.043 *
Serum LPS activity	Dry	26.58	0.918	25.85	0.916	0.0135	0.016	0.399
Glutamine	Dry	1.60	0.952	1.06	0.957	0.0171	0.023	0.494
Hyodeoxycholate	Dry	5.79	0.832	5.76	0.763	0.0043	0.024	0.864
Androsterone sulfate	Wet	17.91	0.211	17.01	0.199	0.0137	0.017	0.421
Epiandrosterone sulfate	Wet	4.95	0.763	4.76	0.689	0.0094	0.022	0.682
Serum LPS activity	Wet	43.07	0.263	42.12	0.259	0.0166	0.018	0.366
**Taxa**	**Metabolite**	**Q**	** *p* **	**Q**	** *p* **	**Intercept**	**Se**	** *p* **
Family *Oxalobacteraceae*	Valine	4.69	0.791	4.66	0.702	−0.0006	0.003	0.870
Genus *Slackia*	Glutamine	7.19	0.303	0.40	0.995	0.0067	0.003	0.048 *

Note: The asterisk (*) represents that the *p* value is less than 0.05. This means that the results show heterogeneity or horizontal pleiotropy.

## Data Availability

The original contributions presented in the study are included in the article/Appendix A. Further inquiries can be directed to the corresponding authors.

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
