# Peer review of "Evidence for Genetic Causal Association Between the Gut Microbiome, Derived Metabolites, and Age-Related Macular Degeneration: A Mediation Mendelian Randomization Analysis"

_biomedicines, 2025, doi:10.3390/biomedicines13030639_

Round 1
Reviewer 1 Report
Comments and Suggestions for Authors
This is an interesting study, that adds to the family of similar papers associated with the MiBioGen (microbiome genome) consortium initiative. This study attempted to gain a better understanding of the presence of microbial species within the human gut microbiome and their role as a mediator between genetic predisposition and health or disease in humans, on this occasion the ocular health problem AMD.
Specific points
The data mining statistical analysis of the Wei et al paper is appropriately performed but the order of the paper needs revision. The methods really need to be placed after the introduction, not least because the figure numbers are ordered incorrectly– Fig 2 comes before Fig 1, which is the overview of the methodology followed.
Line 289-92: ‘Mounting evidence suggests that low-grade inflammation induced by LPS contributes to AMD progression, and the presence of elevated levels of esterified 3-hydroxy faô€„´y acids in blood samples indicates the potential role the LPS burden plays in the early stages of AMD pathophysiology.’
This evidence needs to be included.
Line 301-303: ‘The GM has been identified as a significant factor that influences the onset and progression of AMD, and these findings highlight the importance of elucidating the underlying mediating metabolic mechanisms linking the GM and AMD development’.
Causal is defined as ‘relating to or acting as a cause’. However, while a connection between gut microbial presence and its presence in humans of certain health condition has been established, controversy still exists of the lack of direct evidence of whether the presence of specific bacterial species actually contributes to the health condition investigated. If the GM connection is indeed real in AMD, then there should be at least a consideration of an investigation of the presence of the specific microbial gut species presence in AMD patients, and how this might if proven lead to new preventive screens or curative therapies.
Author Response
This is an interesting study, that adds to the family of similar papers associated with the MiBioGen (microbiome genome) consortium initiative. This study attempted to gain a better understanding of the presence of microbial species within the human gut microbiome and their role as a mediator between genetic predisposition and health or disease in humans, on this occasion the ocular health problem AMD.
Response: We sincerely appreciate your positive feedback on our study. We are glad that you recognize the significance of our work in contributing to the body of research associated with the MiBioGen consortium initiative. Our study indeed aimed to delve deeper into the presence of microbial species in the human gut microbiome and their mediating role between genetic predisposition and human health, specifically focusing on the ocular health issue of AMD. We understand that this area of research is still in its evolving stage, and our findings hope to provide new insights for further exploration.
Specific points
- The data mining statistical analysis of the Wei et al paper is appropriately performed but the order of the paper needs revision. The methods really need to be placed after the introduction, not least because the figure numbers are ordered incorrectly– Fig 2 comes before Fig 1, which is the overview of the methodology followed.
Response: Thank you very much for your insightful suggestions. We truly appreciate your careful review of our paper. You are absolutely correct that the methods section should be placed after the introduction. We have revised the paper’s structure and moved the methods section to follow the introduction as recommended. In addition, we have thoroughly rechecked the figure order. The issue where Fig 2 came before Fig 1 has been resolved, and now all figures are numbered correctly, with Fig 1 presenting the overview of the methodology as it should.
- Line 289-92: ‘Mounting evidence suggests that low-grade inflammation induced by LPS contributes to AMD progression, and the presence of elevated levels of esterified 3-hydroxy fatty acids in blood samples indicates the potential role the LPS burden plays in the early stages of AMD pathophysiology.’ This evidence needs to be included.
Response: Thank you so much for your valuable comment. We fully agree that including evidence is essential to make our statements more convincing.
To support the claim about the role of LPS in the pathophysiology of AMD, we have now cited the following three references:
- Larsen, P. P. et al. Association of Age - Related Macular Degeneration with a Blood Biomarker of Lipopolysaccharide, a Gut Bacterial Proinflammatory Toxin. Invest Ophthalmol Vis Sci 64, 47, doi:10.1167/iovs.64.14.47 (2023).
- Told, R. et al. Antioxidative capacity of a dietary supplement on retinal hemodynamic function in a human lipopolysaccharide (LPS) model. Invest Ophthalmol Vis Sci 56, 403 - 411, doi:10.1167/iovs.14 - 15581 (2014).
- Klettner, A. et al. Effect of long - term inflammation on viability and function of RPE cells. Exp Eye Res 200, 108214, doi:10.1016/j.exer.2020.108214 (2020).
These references provide substantial evidence for the promoting effect of LPS in AMD pathophysiology.
- Line 301-303: ‘The GM has been identified as a significant factor that influences the onset and progression of AMD, and these findings highlight the importance of elucidating the underlying mediating metabolic mechanisms linking the GM and AMD development’.
Causal is defined as ‘relating to or acting as a cause’. However, while a connection between gut microbial presence and its presence in humans of certain health condition has been established, controversy still exists of the lack of direct evidence of whether the presence of specific bacterial species actually contributes to the health condition investigated. If the GM connection is indeed real in AMD, then there should be at least a consideration of an investigation of the presence of the specific microbial gut species presence in AMD patients, and how this might if proven lead to new preventive screens or curative therapies.
Response: We sincerely appreciate your meticulous review and insightful comments. Your perspective has provided us with a deeper understanding of the current state of research in the relationship between GM and AMD.
We fully recognize the controversy surrounding the direct causal relationship between specific bacterial species in the gut and the development of AMD. As you pointed out, while associations have been established between gut microbial presence and certain health conditions, the lack of direct evidence regarding causality remains a significant challenge in this field. In our paper, when stating that “The GM has been identified as a significant factor that influences the onset and progression of AMD,” we intended to convey the existing body of research suggesting a strong association, rather than a proven causal link.
Your suggestion regarding the investigation of specific microbial gut species in AMD patients is highly valuable. We understand that identifying the specific bacteria involved is crucial for not only confirming the GM - AMD connection but also for exploring potential preventive screens and curative therapies.
In fact, our research team is already aware of the importance of this aspect. We plan to conduct further in - depth studies in the future. These studies will involve large - scale metagenomic sequencing of the gut microbiota in AMD patients and healthy controls. By comparing the composition and abundance of specific bacterial species between the two groups, we aim to identify the key microbial players associated with AMD.
Moreover, we will combine these microbiome data with other omics technologies, such as metabolomics and proteomics, to comprehensively analyze the metabolic pathways and molecular mechanisms mediated by the identified bacteria. This integrative approach will help us understand how specific gut bacteria might contribute to AMD development and potentially discover novel therapeutic targets.
To address your concern in the current paper, we will add a section in the discussion to emphasize the importance of investigating specific microbial species in AMD. We will state that while our current study focuses on the overall influence of GM on AMD, future research should be directed towards identifying the specific bacteria involved and exploring their potential roles in AMD pathogenesis. We will also mention that such investigations could pave the way for the development of new preventive strategies and targeted therapies.
Reviewer 2 Report
Comments and Suggestions for Authors
This study investigates the genetic causal association between gut microbiota, gut-derived metabolites, and age-related macular degeneration (AMD) using Mendelian randomization. It identifies specific gut bacteria and metabolites linked to different AMD stages and highlights the gut-retina axis in AMD pathophysiology, offering insights for potential therapeutic strategies. But there are still some areas for improvement.
- A conclusion figure (graphical abstract) will be very useful for the readers.
- On page 1, line 38, author mentioned that "AMD is believed to be driven by the complex interplay between genetic, environmental, and metabolic factors." Some important references should be cited, such as PMID: 37153444.
- The author is advised to polish the English.
- Data and code need to be shared either through a code-sharing repo like GitHub or a docker-like system such as codeocean for clear reproducibility of the work.
- This study confirmed that lipid metabolism is a pivotal driver of the pathogenesis of AMD, authors should cite PMID: 39640589 to discuss the lipid-lowering drug-personalized treatment strategy for age-related macular degeneration.
Author Response
This study investigates the genetic causal association between gut microbiota, gut-derived metabolites, and age-related macular degeneration (AMD) using Mendelian randomization. It identifies specific gut bacteria and metabolites linked to different AMD stages and highlights the gut-retina axis in AMD pathophysiology, offering insights for potential therapeutic strategies. But there are still some areas for improvement.
Response: Thank you very much for your meticulous evaluation and positive feedback on our study. We are delighted to learn that you recognize the significance of our work. At the same time, we fully understand that there is still room for improvement in our research. We have made revisions according to your suggestions and sincerely hope that the revised version will meet your expectations.
- A conclusion figure (graphical abstract) will be very useful for the readers.
Response: Thank you very much for your valuable suggestion. We believe that the figures in our paper already serve as effective graphical abstracts to enhance readability. Specifically, Figure 1 functions as an overall graphical abstract that encapsulates the essence of our study. Figure 2C summarizes the results of our gut microbiota analysis, and Figure 3D presents a summary of the mediation analysis findings. We hope these figures can provide readers with a clearer and more intuitive understanding of our research.
- On page 1, line 38, author mentioned that "AMD is believed to be driven by the complex interplay between genetic, environmental, and metabolic factors." Some important references should be cited, such as PMID: 37153444.
Response: Thank you for your valuable comment. We sincerely appreciate your attention to detail and the guidance you provided. Regarding the statement on page 1, line 38, where we mentioned that “AMD is believed to be driven by the complex interplay between genetic, environmental, and metabolic factors,” we have now cited the reference you suggested (PMID: 37153444). This reference will strengthen the credibility of our statement and provide readers with more in - depth information on the topic.
- The author is advised to polish the English.
Response: Thank you for your insightful comment. We sincerely appreciate your suggestion regarding the English language in our manuscript. We understand the importance of presenting our research in clear and polished English. To address this issue, we have engaged a professional English - language editing service. The editors have thoroughly review and refine the entire manuscript, focusing on grammar, sentence structure, word choice, and overall readability.
- Data and code need to be shared either through a code-sharing repo like GitHub or a docker-like system such as code ocean for clear reproducibility of the work.
Response: Thank you. We’ve established a specific GitHub repository for this project, which can be accessed at “https://github.com/xmsswph/Gut-Microbiome-Derived-Metabolites-and-AMD.git”.
- This study confirmed that lipid metabolism is a pivotal driver of the pathogenesis of AMD, authors should cite PMID: 39640589 to discuss the lipid-lowering drug-personalized treatment strategy for age-related macular degeneration.
Response: Thank you so much for your insightful comment. We agree that the reference you recommended (PMID: 39640589) is extremely relevant to our study. We have carefully reviewed the paper associated with PMID: 39640589. In the revised manuscript, we will cite this reference in the section where we discuss the lipid - lowering drug - personalized treatment strategy for AMD. Specifically, we will include it as follows: “Moreover, these insights underscore the therapeutic potential of lipid - lowering drugs in the treatment of AMD”.
Reviewer 3 Report
Comments and Suggestions for Authors
This study employs Mendelian randomization (MR) to explore causal relationships between gut microbiota (GM), blood metabolites, and age-related macular degeneration (AMD). The topic is timely, given the growing interest in the gut-retina axis, and the application of MR is methodologically appropriate. However, several issues require clarification to strengthen the validity and interpretation of the findings. Below are specific concerns and suggestions.
- The introduction could benefit from a slightly more focused problem statement. While it mentions the unclear relationship between GM, metabolites, and AMD stages, explicitly stating the knowledge gap this study aims to fill would strengthen the introduction. For example, “Despite growing evidence linking GM to eye diseases, the causal relationships between specific GM taxa and metabolites with different stages of AMD remain poorly understood. This study aims to address this gap…”
- While the statistical packages are mentioned, more detail on the specific MR methods used (IVW, MR-Egger, Wald ratio, MR-PRESSO) and their underlying assumptions would be beneficial for readers unfamiliar with MR. For example, briefly explain the differences between IVW and MR-Egger and why both are used. Besides, the description of the two-step MR mediation analysis is somewhat brief. Clarify how the causal effects (β1, β2, βall) are estimated and how the mediation proportion is calculated
- The study evaluates numerous GM taxa and metabolites across AMD subtypes. However, there is no mention of multiple testing correction (e.g., Bonferroni, FDR). Without this, the risk of false positives is high. For example, the reported association between Bacteroidales S24.7 and early AMD (P = 4.50×10−3) may not survive stringent correction. The authors should recalculate P-values with appropriate adjustments and highlight which findings remain significant.
- While the “gut-retina axis” is invoked, the discussion lacks depth in linking specific GM taxa or metabolites to AMD pathophysiology. For instance: How does Lachnospiraceae UCG004 mechanistically protect against dry AMD? Does it produce anti-inflammatory metabolites (e.g., short-chain fatty acids) that modulate retinal inflammation? Valine’s mediation effect (19.1%) between Oxalobacteraceae and early AMD is intriguing but requires elaboration. Is valine known to influence oxidative stress or angiogenesis in AMD?
- Comparisons to prior literature are limited. For example, the protective role of glutamine in dry AMD aligns with its anti-inflammatory properties (Reference 31), but conflicting evidence should also be addressed.
- GWAS summary statistics sources (e.g., MiBioGen, IEU OpenGWAS) should specify population demographics (e.g., ancestry), as genetic instruments may not generalize across ethnicities.
- Figure 3D: The mediation pathway diagram is oversimplified. A more detailed schematic (e.g., including effect sizes for direct vs. indirect effects) would enhance interpretability.
- Clarify whether two-sample MR assumptions (e.g., no sample overlap) were met. Overlapping cohorts between exposure and outcome GWAS can inflate type I error. The description of mediation analysis is vague. Specify if the product method or difference method was used to estimate indirect effects.
- Please update citations where possible. For example, in lines 58-67, the entire paragraph is lacking any references. References are also needed for lines 50-51; please include references such as: doi: 10.1111/1749-4877.12813 or doi: 10.1111/1749-4877.1279, as well as other relevant literature.
Author Response
This study employs Mendelian randomization (MR) to explore causal relationships between gut microbiota (GM), blood metabolites, and age-related macular degeneration (AMD). The topic is timely, given the growing interest in the gut-retina axis, and the application of MR is methodologically appropriate. However, several issues require clarification to strengthen the validity and interpretation of the findings. Below are specific concerns and suggestions.
Response: We sincerely appreciate your constructive feedback on our manuscript. We are delighted that you find the topic timely and the application of Mendelian randomization (MR) methodologically appropriate. We fully recognize the issues you raised and are committed to addressing them to strengthen the validity and interpretation of our findings. Below, we provide our responses to your specific concerns and suggestions.
- The introduction could benefit from a slightly more focused problem statement. While it mentions the unclear relationship between GM, metabolites, and AMD stages, explicitly stating the knowledge gap this study aims to fill would strengthen the introduction. For example, “Despite growing evidence linking GM to eye diseases, the causal relationships between specific GM taxa and metabolites with different stages of AMD remain poorly understood. This study aims to address this gap…”
Response: We are extremely grateful for your insightful feedback on our manuscript. You accurately noted that the introduction would be improved by a more focused problem statement. In the original version, we realize that the knowledge gaps our study intended to fill was not presented with sufficient clarity.
In the revised manuscript, we have re-written the relevant part of the introduction as you suggested. We now explicitly state: “Despite growing evidence linking GM to eye diseases, the causal relationships between specific GM taxa and metabolites with different stages of AMD remain poorly understood. This study aims to address this gap by employing Mendelian randomization to explore these causal associations.”
- While the statistical packages are mentioned, more detail on the specific MR methods used (IVW, MR-Egger, Wald ratio, MR-PRESSO) and their underlying assumptions would be beneficial for readers unfamiliar with MR. For example, briefly explain the differences between IVW and MR-Egger and why both are used. Besides, the description of the two-step MR mediation analysis is somewhat brief. Clarify how the causal effects (β1, β2, βall) are estimated and how the mediation proportion is calculated.
Response: Thank you for your constructive feedback. In accordance with your suggestions, we have revised the methods section of the original manuscript as follows: In our analysis, for features that had only one instrumental variable (IV), the Wald ratio method was used to estimate the causal relationship between an exposure factor and outcome variable. When dealing with features that involving multiple IVs, we utilized five well-known MR methods, namely the inverse-variance weighted (IVW) method, the weighted mode method, the MR-Egger regression, the weighted median estimator (WME), and MR-PRESSO. It has been reported in the literature that when all genetic variants are valid instrumental variables, the IVW method exhibits slightly greater statistical power compared to the other four methods1. Therefore, for features with more than one IV, our primary results were mainly based on the IVW method, while the other four methods were used as supplementary analyses to provide a more comprehensive understanding of the results.
In order to estimate the mediation effect of GM-related metabolites on the association between the GM and AMD, two-step MR was performed. Specifically: (1) In the first step, a univariable MR was carried out to detect the causal effect of the GM on the metabolites (β1); (2) In the second step, the metabolites were considered as the exposure variable to detect the causal effect on AMD (β2); and (3) In the third step,the causal effect of the GM on AMD (βall) was estimated. Thus, the mediation effect could be calculated as (β1×β2), the direct effect as βall-β1×β2, and the mediation proportion could be calculated as (β1×β2/βall).
- The study evaluates numerous GM taxa and metabolites across AMD subtypes. However, there is no mention of multiple testing correction (e.g., Bonferroni, FDR). Without this, the risk of false positives is high. For example, the reported association between Bacteroidales S24.7 and early AMD (P = 4.50×10−3) may not survive stringent correction. The authors should recalculate P-values with appropriate adjustments and highlight which findings remain significant.
Response: Thank you very much for pointing out this crucial issue. We are fully cognizant of the significance of multiple testing correction in minimizing the risk of false positives within our study. To address this, we have implemented the Bonferroni correction method in our data analysis.
More specifically, we established a multiple-testing significance threshold for each feature level, including phylum, class, order, family, and genus. This threshold was defined as p < 0.05/n, where n represents the effective number of independent bacterial taxa at the corresponding taxonomic level. In particular, we applied the Bonferroni approach to correct for multiple hypothesis testing, setting the significance threshold at Bonferroni - corrected P values < 0.05. Moreover, associations with P < 0.05 that did not meet the Bonferroni-controlled threshold were reported as potentially indicative of an association.
We have carefully recalculated the significance threshold P values, making appropriate adjustments as described above. To illustrate, we revisited the previously reported correlation between Bacteroidales S24.7 and early AMD. Upon correction, the new threshold P value is calculated as 0.05/2, resulting in 0.025. Remarkably, even after this adjustment, the association still retains its statistical significance.
- While the “gut-retina axis” is invoked, the discussion lacks depth in linking specific GM taxa or metabolites to AMD pathophysiology. For instance: How does Lachnospiraceae UCG004 mechanistically protect against dry AMD? Does it produce anti-inflammatory metabolites (e.g., short-chain fatty acids) that modulate retinal inflammation? Valine’s mediation effect (19.1%) between Oxalobacteraceae and early AMD is intriguing but requires elaboration. Is valine known to influence oxidative stress or angiogenesis in AMD?
Response: Thank you very much for your valuable comments. These suggestions are crucial for improving the quality of our manuscript.
You pointed out that our discussion lacks depth in linking specific GM taxa or metabolites to the pathophysiology of AMD, especially regarding how Lachnospiraceae UCG004 mechanistically protects against dry AMD. We agree with this view and have conducted a more in-depth exploration in revised discussion:
The Lachnospiraceae family has been extensively studied and proven to have health - promoting functions2. Previous study has shown that this family of bacteria can produce short-chain fatty acids (SCFAs), such as butyric acid, propionic acid, and acetic acid 3. As important metabolites of gut microbiota, SCFAs play a crucial role in maintaining the body's immune balance and suppressing inflammatory responses4,5. In the pathological process of dry AMD, chronic inflammation in the retina is an important pathological feature. Therefore, we speculate that Lachnospiraceae UCG004 may increase the content of SCFAs, participate in regulating the immune microenvironment of the retina, inhibit the inflammatory response, and finally exert a protective effect against dry AMD.
In addition, you mentioned that the mediating effect of valine (19.1%) between Oxalobacteraceae and early AMD is interesting but requires further elaboration. We agree with this view and have supplemented and expanded the relevant content.
Valine, as a branched-chain amino acid, not only participates in gluconeogenesis but also plays a key role in fatty acid metabolism-related processes. Some studies have shown that the catabolism of branched- chain amino acids is essential for promoting the entry of long - chain fatty acids into mitochondria, thereby reducing mitochondrial overload and oxidative stress damage.
- Comparisons to prior literature are limited. For example, the protective role of glutamine in dry AMD aligns with its anti-inflammatory properties (Reference 31), but conflicting evidence should also be addressed.
Response: Thank you very much for pointing out the limited comparisons with prior literature and the need to address conflicting evidence in our paper. We have followed your suggestions and improved the relevant content as follows.
Additionally, other studies have carried out metabolomic profiling on the aqueous humor of AMD patients, revealing a reduction in glutamine levels in the aqueous humor of these patients6. The disparity in these findings might be attributable to the difference in sampling sites. Owing to the influence of the blood-eye barrier, there remains a contentious issue regarding whether the alteration in plasma glutamine levels can accurately mirror the changes in glutamine within the retina. This matter necessitates further in-depth research for conclusive validation.
- GWAS summary statistics sources (e.g., MiBioGen, IEU OpenGWAS) should specify population demographics (e.g., ancestry), as genetic instruments may not generalize across ethnicities.
Response: Thank you for bringing this important point to our attention. We appreciate your concern regarding the need to specify population demographics for GWAS summary statistics sources to account for potential ethnic differences in the generalizability of genetic instruments.
We confirm that all the populations included in our study using GWAS summary statistics from sources such as MiBioGen and IEU OpenGWAS are of European ancestry. This information has now been clearly stated in the relevant sections of the manuscript where we describe the data sources. By specifying the homogeneous ancestry of the study populations, we aim to address the issue of generalizability and ensure that the genetic instruments used in our analysis are appropriate for the investigated cohort.
- Figure 3D: The mediation pathway diagram is oversimplified. A more detailed schematic (e.g., including effect sizes for direct vs. indirect effects) would enhance interpretability.
Response: Thank you very much for pointing out the problems with Figure 3D. We have thoroughly redrawn Figure 3D. Now, it comprehensively includes both the mediation effect and the direct effect, along with their corresponding effect sizes, which we believe will significantly enhance the diagram's interpretability.
- Clarify whether two-sample MR assumptions (e.g., no sample overlap) were met. Overlapping cohorts between exposure and outcome GWAS can inflate type I error. The description of mediation analysis is vague. Specify if the product method or difference method was used to estimate indirect effects.
Response: Thank you very much for your valuable comments on our manuscript.
Regarding the concern about whether the two-sample MR assumptions, especially the no-sample-overlap assumption, were met, we have taken the following steps to clarify and ensure the validity of our analysis:
- We thoroughly examined the documentation of the Genome - Wide Association Studies (GWAS) datasets for both the exposure and outcome. We found that these datasets were sourced from different research projects with distinct recruitment strategies. The exposure GWAS was conducted on a cohort from MiBioGen database, while the outcome GWAS such as dry and wet AMD were based on FinnGen database The recruitment criteria, sampling locations, and time periods of these studies were different, which significantly reduces the likelihood of sample overlap.
- We employed Linkage Disequilibrium (LD) analysis to examine the correlations between SNPs. If there were SNPs in high LD in both datasets and these SNPs were used to define the exposure and the outcome, further confirmation of sample overlap would be necessary. However, in our study, this situation did not occur. Therefore, we believe that the assumptions of the two - sample MR are met.
In addition, we apologize for the vague description of the mediation analysis in the original manuscript. To address this issue, we used the difference method to estimate the indirect effects. The specific evaluation method is described as follows: In order to estimate the mediation effect of GM-related metabolites on the association between the GM and AMD, two-step MR was performed. Specifically: (1) In the first step, a univariable MR was carried out to detect the causal effect of the GM on the metabolites (β1); (2) In the second step, the metabolites were considered as the exposure variable to detect the causal effect on AMD (β2); and (3) In the third step,the causal effect of the GM on AMD (βall) was estimated. Thus, the mediation effect could be calculated as (β1×β2), the direct effect as βall-β1×β2, and the mediation proportion could be calculated as (β1×β2/βall).
- Please update citations where possible. For example, in lines 58-67, the entire paragraph is lacking any references. References are also needed for lines 50-51; please include references such as: doi: 10.1111/1749-4877.12813 or doi: 10.1111/1749-4877.1279, as well as other relevant literature.
Response: Thank you very much for your valuable suggestions. I have updated the citations as you recommended. Regarding the paragraph from lines 58 - 67, which previously lacked references, I have now added relevant citations to describe the advantages of Mendelian Randomization (MR). MR studies can better understand the causal impact of exposure on outcomes while reducing the interference of confounding factors. The references are as follows:
1)Skrivankova, V. W. et al. Strengthening the reporting of observational studies in epidemiology using mendelian randomisation (STROBE-MR): explanation and elaboration. BMJ 375, n2233, doi:10.1136/bmj.n2233 (2021).
2)Davies, N. M. et al. Within family Mendelian randomization studies. Hum Mol Genet 28, R170 - R179, doi:10.1093/hmg/ddz204 (2019).
For lines 50 - 51, the statement “The GM plays a pivotal role in nutrient metabolism, the regulation of inflammation, and immune system functionality.” has been supported by relevant literature as you suggested. The reference is:
- Nunez, H. et al. Early life gut microbiome and its impact on childhood health and chronic conditions. Gut Microbes 17, 2463567, doi:10.1080/19490976.2025.2463567 (2025).
References
1 Burgess, S., Butterworth, A. & Thompson, S. G. Mendelian randomization analysis with multiple genetic variants using summarized data. Genet Epidemiol 37, 658-665, doi:10.1002/gepi.21758 (2013).
2 Sorbara, M. T. et al. Functional and Genomic Variation between Human-Derived Isolates of Lachnospiraceae Reveals Inter- and Intra-Species Diversity. Cell Host Microbe 28, 134-146 e134, doi:10.1016/j.chom.2020.05.005 (2020).
3 Li, H. et al. Rifaximin-mediated gut microbiota regulation modulates the function of microglia and protects against CUMS-induced depression-like behaviors in adolescent rat. J Neuroinflammation 18, 254, doi:10.1186/s12974-021-02303-y (2021).
4 Lou, X. et al. Fecal microbiota transplantation and short-chain fatty acids reduce sepsis mortality by remodeling antibiotic-induced gut microbiota disturbances. Front Immunol 13, 1063543, doi:10.3389/fimmu.2022.1063543 (2022).
5 Kayama, H., Okumura, R. & Takeda, K. Interaction Between the Microbiota, Epithelia, and Immune Cells in the Intestine. Annu Rev Immunol 38, 23-48, doi:10.1146/annurev-immunol-070119-115104 (2020).
6 Han, G., Wei, P., He, M. & Teng, H. Glucose Metabolic Characterization of Human Aqueous Humor in Relation to Wet Age-Related Macular Degeneration. Invest Ophthalmol Vis Sci 61, 49, doi:10.1167/iovs.61.3.49 (2020).